# Human Multi-Activities Classification Using mmWave Radar: Feature Fusion in Time-Domain and PCANet

**DOI:** 10.3390/s24165450

**Published:** 2024-08-22

**Authors:** Yier Lin, Haobo Li, Daniele Faccio

**Affiliations:** 1School of Electronic Science and Engineering, University of Electronic Science and Technology of China, Chengdu 611731, China; yierlin@foxmail.com; 2Beijing Vocational College of Transport, Beijing 102618, China; 3Department of Biomedical Engineering, School of Science and Engineering, University of Dundee, Dundee DD1 4HN, UK; hli005@dundee.ac.uk; 4Extreme Light Group, School of Physics & Astronomy, University of Glasgow, Glasgow G12 8QQ, UK

**Keywords:** human activity recognition, CNN-BiLSTM, mmWave, feature fusion, point cloud

## Abstract

This study introduces an innovative approach by incorporating statistical offset features, range profiles, time–frequency analyses, and azimuth–range–time characteristics to effectively identify various human daily activities. Our technique utilizes nine feature vectors consisting of six statistical offset features and three principal component analysis network (PCANet) fusion attributes. These statistical offset features are derived from combined elevation and azimuth data, considering their spatial angle relationships. The fusion attributes are generated through concurrent 1D networks using CNN-BiLSTM. The process begins with the temporal fusion of 3D range–azimuth–time data, followed by PCANet integration. Subsequently, a conventional classification model is employed to categorize a range of actions. Our methodology was tested with 21,000 samples across fourteen categories of human daily activities, demonstrating the effectiveness of our proposed solution. The experimental outcomes highlight the superior robustness of our method, particularly when using the Margenau–Hill Spectrogram for time–frequency analysis. When employing a random forest classifier, our approach outperformed other classifiers in terms of classification efficacy, achieving an average sensitivity, precision, F1, specificity, and accuracy of 98.25%, 98.25%, 98.25%, 99.87%, and 99.75%, respectively.

## 1. Introduction

The World Health Organization has noted a significant growth in the global population of individuals aged 60 and above, from 1 billion in 2019 to an anticipated 1.4 billion by 2030 and further to 2.1 billion by 2050 [1]. Specifically, in China, the elderly population is expected to reach 402 million, making up 28% of the country’s total population by 2040, due to lower birth rates and increased longevity [2]. This demographic shift has led to a rise in fall-induced injuries among the elderly, which is a leading cause of discomfort, disability, loss of independence, and early mortality. Statistics indicate that 28–35% of individuals aged 65 and above fall each year, which is a figure that increases to 32–42% for those aged over 70 [3]. As such, the remote monitoring of senior activities is becoming increasingly important.

Techniques for recognizing human activities fall into three categories: those that rely on wearable devices, camera systems, and sensor technologies [4]. Unlike wearables [5], camera- [6] and sensor-based methods offer the advantage of being fixed in specific indoor locations, eliminating the need for constant personal wear and thus better serving the home-bound elderly through remote monitoring while also addressing privacy concerns. Sensor-based technologies, in particular, utilize radio frequency signals’ phase and amplitude without generating clear images, significantly reducing privacy violations compared to traditional methods.

Over the past decades, research in human activity recognition has explored various sensors. K. Chetty demonstrated the potential of using passive bistatic radar with WiFi for covertly detecting movement behind walls, enhancing signal clarity with the CLEAN method [7]. V. Kilaru explored identifying stationary individuals through walls using a Gaussian mixture model [8]. In our previous research, we developed a software-defined Doppler radar sensor for activity classification and employed various time–frequency image features, including the Choi-Williams and Margenau-Hill Spectrograms [9]. We proposed iterative convolutional neural networks with random forests [10] to accurately categorize several activities and individuals using FMCW radar, achieving successful activity recognition in diverse settings. Unfortunately, using the entire autocorrelation signal as input for the iterative convolutional neural networks with random forests led to excessive computational workload and frequent out-of-memory errors.

Thanks to numerous scattering centers, often called point clouds, mmWave radars are liable to provide high resolution. Because of their low cost and ease of use, mmWave radars have been gaining popularity. The team of A. Sengupta detected and tracked real-time human skeletons using mmWave radar in [11], while the team of S. An proposed a millimeter-based assistive rehabilitation system and a 3D multi-model human pose estimation dataset in [12,13]. Unfortunately, training fine-grained, accurate activity classifiers is challenging, as low-cost mmWave radar systems produce sparse, non-uniform point clouds.

This article introduces a cutting-edge classification algorithm designed to address the shortcomings of sparse and non-uniform point clouds generated by economical mmWave radar systems, thereby enhancing their applicability in practical scenarios. Our innovative solution boosts performance through the integration of statistical offset measures, range profiles, time–frequency analyses, and azimuth–range–time evaluations. It features nine distinct feature vectors composed of six statistical offset measures and three principal component analysis network (PCANet) fusion features derived from range profiles, time–frequency analyses, and azimuth–range–time imagery. These statistical offset measures are derived from I/Q data, which are harmonized through an angle relationship incorporating elevation and azimuth details. The fusion process involves channeling elevation and azimuth PCANet inputs through simultaneous 1D CNN-BiLSTM networks. This method prioritizes temporal fusion in the CNN-BiLSTM architecture for 3D range–azimuth–time data, followed by PCANet integration, enhancing the quality of fused features across the board and thereby improving overall classification accuracy.

The key contributions of this research are highlighted as follows:(1)The introduction of an original method for classifying fourteen types of human daily activities, leveraging statistical offset measures, range profiles, time–frequency analyses, and azimuth–range–time data.(2)Pioneering the use of the CNN-BiLSTM framework for fusing 3D range–azimuth–time information.(3)Recommendation of the Margenau–Hill Spectrogram (MHS) for optimal feature quality and minimal feature count, which has been validated by analysis of four time–frequency methods.

In this paper, scalars and vectors are denoted by lowercase letters *x* and bold lowercase letters x, whereas = denotes the equal operator.

The remainder of this paper is structured as follows. Section 2 introduces the related works. Section 3 describes our methodology details using radar formula, feature source, feature fusion structure, and the framework of our method. Section 4 describes the experimental environment and the recording process of the measurement data. Our algorithm performance outcomes are illustrated with numerical results from the combinations classification of human daily action, which are also in Section 4. Future works and conclusions are drawn in Section 5.

## 2. Related Work

Numerous studies have focused on human activity recognition using millimeter wave (mmWave) technology. For instance, A. Pearce et al. [14] provided an in-depth review of the literature on multi-object tracking and sensing using short-range mmWave radar, while J. Zhang et al. [15] summarized the work on mmWave-based human sensing, and H. D. Mafukidze et al. [16] explored advancements in mmWave radar applications, offering ten public datasets for benchmarking in target detection, recognition, and classification.

Research employing feature-level fusion technology for human activity recognition includes a study in [17], which calculated mean Doppler shifts alongside micro-Doppler, statistical, time, and frequency domain features to feed into a support vector machine, achieving between 98.31% and 98.54% accuracy in classifying activities such as falling, walking, standing, sitting, and picking. E. Cardillo et al. [18] used range-Doppler and micro-Doppler features to classify different users’ gestures. Given the challenges posed by sparse and irregular point clouds from budget-friendly mmWave radar, data enhancement techniques have become crucial for achieving high accuracy. A method proposed in [19] created samples with varied angles, distances, and velocities of human motion specifically for mmWave-based activity recognition. Another study in [20] utilized radar image classification and data augmentation, along with principal component analysis, to distinguish six activities, achieving 95.30% accuracy with the convolutional neural network (CNN) algorithm. E. Kurto$\overset{ˇ}{g}$lu et al. [21] exploited an approach that utilized multiple radio frequency data domain representations, including range-Doppler, time–frequency, and range–angle, for the sequential classification of fifteen American Sign Language words and three gross motor activities, achieving a detection rate of 98.9% and a classification accuracy of 92%.

Beyond feature-level fusion, autoencoders are widely used in mmWave signal analysis for activity recognition. The mmFall system in [22], utilizing a hybrid variational RNN autoencoder, reported a 98% success rate in detecting falls from 50 instances with only two false alarms. R. Mehta et al. [23] conducted a comparative study on extracting features through a convolutional variational autoencoder, showing the highest classification accuracy of 81.23% for four activities with the Sup-EnLevel-LSTM method.

Point cloud neural network technologies also play a pivotal role in recognizing human activities through mmWave signals. A real-time system in [24], using the PointNet model, demonstrated 99.5% accuracy in identifying five activities. The m-Activity system in [25] filtered human movements from background noise before classification with a specially designed lightweight neural network, resulting in 93.25% offline and 91.52% real-time accuracy for five activities. G. Lee and J. Kim leveraged spatial–temporal domain information for activity recognition using graph neural networks and a pre-trained model on point cloud and Kinect data [26], with their MTGEA model [27] achieving 98.14% accuracy in classifying various activities with mmWave and skeleton data.

Moreover, long short-term memory (LSTM) and CNN technologies have been essential in processing point clouds. C. Kittiyanpunya and team achieved 99.50% accuracy in classifying six activities using LSTM networks with 1D point clouds and Doppler velocity as inputs [28]. An end-to-end learning approach in [29] transformed each point cloud frame into 3D data for a custom CNN model, achieving a recall rate of 0.881. S. Khunteta et al. [30] showcased a technique where CNN extracted features from radio frequency images, followed by LSTM analyzing these features over time, reaching a peak accuracy of 94.7% for eleven activities. RadHAR in [31] utilized a time sliding window to process sparse point clouds into a voxelized format for CNN-BiLSTM classification, achieving 90.47% accuracy. Lastly, DVCNN in [32] improved data richness through radar rotation symmetry, employing a dual-view CNN to recognize activities, attaining accuracies of 97.61% and 98% for fall detection and a range of activities, respectively.

## 3. Methodology

This section includes four subsections. The first one introduces the standard formula of mmWave radar [33] for context. The second one displays where the information comes from. The third one introduces the neural network structure for obtaining the fusion features. The last one presents the framework of our approach.

### 3.1. Radar Formula

In this article, mmWave radar with chirps was applied in human activity recognition. The transmission frequency was linearly increased over time through the transmit antenna. The chirp can be a formulated as follows:(1)$$S\left(t\right)=e^{j2\pi t(f_{c}+\frac{Bt}{2T_{c}}t)},\;0\leq t\leq T_{c}$$
where *B*, $f_{c}$, and $T_{c}$ denote the bandwidth, carrier frequency, and chirp duration.

The time delay of the received signal has the following formula:(2)$$\begin{array}{c} \tau =2(R+vt)/c \end{array},$$
where *R*, *v*, and *c* denote the target range, the target velocity, and the light speed.

A radar frame is a consecutive chirp sequence, which is likely to be reshaped to a two-dimensional waveform across fast and slow time dimensions. There are *M* chirps with a sampling period $T_{rep}$ (slow time dimension) in a frame, and each chirp has *N* points (fast time dimension) with $f_{s}$ as the sampling rate. Hence, the intermediate frequency across fast and slow time dimensions together can be approximated as
(3)$$\begin{array}{c} d\left(n,m\right)\approx e^{j2\pi [\frac{(f_{r}+f_{d})n}{f_{s}}+f_{d}mT_{rep}+\frac{2f_{c}R}{c}]} \end{array},$$
where *n* and *m* denote the index of fast and slow time samples, respectively. The information can be extracted by fast Fourier transform (FFT) along the fast and slow temporal dimensions. The range frequency $f_{r}$ and Doppler $f_{d}$ frequencies can be expressed as
(4)$$\begin{array}{c} f_{r}=\frac{2BR}{cT_{c}} \end{array},$$
(5)$$\begin{array}{c} f_{d}=2f_{c}v/c \end{array}.$$

Thanks to the received signal of each antenna having a different phase, a linear antenna array can estimate the target’s azimuth. The phase shift between received signals from two adjacent antennas can be expressed as
(6)$$\begin{array}{c} \Delta \phi =2\pi \frac{d\sin\theta }{\lambda } \end{array},$$
where θ denotes the target azimuth, while *d* denotes the distance between adjacent antennas. $\lambda =c/f_{c}$ denotes the base wavelength of the transmitted chirp.

For *I* number of targets, the three-dimensional intermediate frequency signal can be approximated as
(7)$$\begin{array}{c} d\left(n,m,l\right)\approx \sum_{i=1}^{I}a_{i}e^{j2\pi [\frac{(f_{rq}+f_{dq})n}{f_{s}}+\frac{ld\sin\theta_{q}}{\lambda }+f_{dq}mT_{rep}+\frac{2f_{c}R_{q}}{c}]} \end{array},$$
where *i* indicates the receiving antenna’s index, while $a_{i}$ denotes the *i*th target’s complex amplitude. The samples of the intermediate frequency signal can be arranged to form a 3D Raw Data Cube across slow time, fast time, and channel dimensions, wherein the FFT can be applied along for velocity, range, and angle estimation.

### 3.2. Feature Source

In this article, the features as the classifiers’ inputs mainly come from four categories of sources, i.e., the offset parameters, range profiles, time–frequency, and range–azimuth–time.

#### 3.2.1. Offset Parameters

Some physical features such as speed and variation rates [34] also are liable to be extracted by the traditional statistic methods using time domain or frequency domain data. This article calculated the offset parameters, including the mean, variance, standard deviation, kurtosis, skewness, and central moment. The mean [35] measures the signal probability distribution central tendency. The variance [36] measures the distance between the signal and its mean. The standard deviation [37] measures the input signal’s variation or dispersion. The kurtosis [38] measures the signal probability distribution tailedness. The skewness [39] measures the asymmetry of the signal probability distribution about its mean. The central moment [40] measures the moment of the signal probability distribution about its mean, and we applied a two-order central moment in the following computation. These six offsets have proven to be effective for classification and were also used in our previous research. Algorithm 1 presents the pseudocode used to compute offsets.
**Algorithm 1** Offsets()**Input:** Elevation, Azimuth**Output:** Offset ParametersSignal=Elevation+Azimuth·$e^{-i\pi /2}$Mean=mean(Signal)Std=std(Signal)Skewness=skewness(Signal)Kurtosis=kurtosis(Signal)Var=var(Signal)Moment=moment(Signal,2)

#### 3.2.2. Range Profiles

The range profile represents the target’s time domain response to a high-range resolution radar pulse. As shown in Figure 1, the range profiles of measured data without desampling, which will be introduced in Section 4.1, show two targets. In this figure, the upper line represents the falling subject, while the lower line represents the standing subject. This is because the mmWave radar provides vertical information, so the range of the falling subject is greater than that of the standing subject when both the base of the standing subject and the radar are at the same horizontal level.

#### 3.2.3. Time-Frequency

Thanks to micro-Doppler signatures, different activities generate uniquely distinctive spectrogram features. Therefore, time-frequency analysis is significant for feature extraction. The most common time-frequency analysis approach is the short-time Fourier transform (STFT) in Figure 2. Moreover, in addition to the STFT, three additional time–frequency distributions (Margenau-Hill Spectrogram [41], Choi-Williams [42], smoothed pseudo Wigner-Ville [43]) and their contributions in the final recognition will be compared in Section 4.

#### 3.2.4. Range–Azimuth–Time

Range–azimuth–time plays a pivotal role in point cloud extraction, particularly in the context of object detection [44]. The range–azimuth dimensions, which are polar coordinates in Figure 3, are often converted into Cartesian coordinates to enhance their interpretability. The following Equations (Equation 8) formulated the coordinate transformation from the range and azimuth domain **[r,θ]** to the Cartesian domain represented by [x,y].
(8)$$\begin{array}{cc} \; & \left\{\begin{array}{c} x=r\cos(\theta )\\ y=r\sin(\theta ) \end{array}\right.,\;\theta \in \left[-\frac{\pi }{2}\;,\frac{\pi }{2}\right]. \end{array}$$

The instantaneous sparse point cloud can be drawn through the constant false alarm rate, as shown in Figure 4. If the time dimension is added to Figure 4, the mmWave radar data can be represented as 4D point clouds.

To reduce the number of features for the final recognition, the time-frequency and range-azimuth images should be analyzed and fused. Orthogonal and linear PCA transform input signals into a new coordinate system to extract main features and reduce the computational complexity [45,46]. The most important variance lies in the first coordinate, while the second most important variance lies in the second. Hence, the first PCA value is the most significant of the whole PCA value, and the following PCA value calculation is based on the former one. Moreover, CNN-BiLSTM has a structure that is BiLSTM. Thereby, the CNN-BiLSTM can fuse the PCANet. We applied PCA to analyze the images and CNN-BiLSTM to fuse the PCANet in this study.

### 3.3. CNN-BiLSTM

CNN-BiLSTM is a network that can reduce and eliminate the network for noise and dimensionality in data using a parallel structure for reducing the time complexity and an attention mechanism for promoting high accuracy via the key representations’ weights redistribution [47,48].

As shown in Figure 5, the values of PCANet are fed into parallel 1D CNN networks, i.e., Stream A and Stream B. The parallel outputs of CNN streams will be multiplied based on the element. After unfolding and flattening, the multiplied sequence will be the input of a bidirectional LSTM (BiLSTM) structure. The BiLSTM structure includes two LSTM networks. The first LSTM network is for forward learning from the previous values, while the second LSTM network is for inverse learning from the upcoming values. The learned information will be combined in the attention layer. The attention layer has four hidden fully connected layers and a softmax, which is used to merge the upstream layer’s output and filter the significant representations out for recognition purposes, i.e., the fusion feature of the inputs. The pseudocode for computing CNN-BiLSTM with K-fold cross-validation is shown in Algorithm 2, and its options are listed in Table 1.The usage of trainNetwork is shown in Figure 6, and its analysis is listed in Table 2.
**Algorithm 2** CNN-BiLSTM()**Input:** FusionInput, KFoldNum**Output:** Fusion FeatureCreate CNN-BiLSTM Layers.Set CNN-BiLSTM Options.$COV=\mathrm{cvpartition}(Label{,}^{\prime }KFold^{\prime },KFoldNum)$;**for** group=1:KFoldNum **do**    Index_Train(group,:)=find(training(COV,group));    Index_Test(group,:)=find(test(COV,group));    Train=InputFusion(Index_Train(group,:),:);    Test=InputFusion(Index_Test(group,:),:);    res=[Train;Test];    **for** i=1:length(unique(Label)) **do**        mid_res=res((res(:,end)==i),:);        mid_size=size(mid_res,1);        mid_tiran=round(1−1/KFoldNum×mid_size);        P_train=[P_train;mid_res(1:mid_tiran,1:end−1)];        T_train=[T_train;mid_res(1:mid_tiran,end)];        P_test=[P_test;mid_res(mid_tiran+1:end,1:end−1)];        T_test=[T_test;mid_res(mid_tiran+1:end,end)];    **end for**    net=trainNetwork(P_train,T_train,Layers,Options);    $T\_sim1\left(group,:\right)=\mathrm{vec}2\mathrm{ind}(\mathrm{predict}{\left(net,P\_train\right)}^{\prime })$;    $T\_sim2\left(group,:\right)=\mathrm{vec}2\mathrm{ind}(\mathrm{predict}{\left(net,P\_test\right)}^{\prime })$;**end for****for** ii=1:group **do**    temp_T_sim1=T_sim1(ii,:);    temp_T_sim2=T_sim2(ii,:);    **for** jj=1:size(temp_T_sim1,2) **do**        Temp(Index_Train(ii,jj))=temp_T_sim1(jj);    **end for**    **for** jj=1:size(temp_T_sim2,2) **do**        Temp(Index_Test(ii,jj))=temp_T_sim2(jj);    **end for**    Fea(ii,:)=Temp;**end for**FuseFeature=mean(Fea,1);

### 3.4. Method Framework

Due to low-cost mmWave radar systems producing sparse, non-uniform point clouds—leading to low-quality feature extraction—training fine-grained, accurate activity classifiers is challenging. To solve this problem, fewer and more high-quality features for final classification are required to boost classification performance. Our novel approach for human activity classification, based on the statistical offset parameters, range profiles, time–frequency, and azimuth–range–time, is presented in Figure 7. Our method has nine fused feature vectors as the input of the classifier.

One of the key aspects of our method is the use of mmWave radar, which is capable of measuring elevation and azimuth information through its scanning method. We leveraged this capability by applying the angle relationship in space to fuse these two types of information, creating fused I/Q data. This data were then used for six statistical feature calculations: mean, variance, standard deviation, skewness, kurtosis, and two-order central moment.

Another one of the key aspects of our method is fusing PCANet images of range profiles and time–frequency. After the CNN-BiLSTM training, two fused vectors of range profiles and time–frequency can be obtained. In this part, we applied the elevation and azimuth data to obtain the PCANet instead of the angle relation fused I/Q data. The elevation and azimuth PCANet values were fed into parallel 1D networks of CNN-BiLSTM. We calculated the fusing PCANet features of images from range profiles or time–frequency using the pseudocode, as shown in Algorithms 3 and 4.
**Algorithm 3** RangeFusion()**Input:** Elevation, Azimuth and Label of Sample Data, KFoldNum**Output:** Fusion Feature of Range Profiles of SamplesCalculate Range Profiles of Elevation and Azimuth respectively.Calculate PCANet of Range Profiles of Samples via SVD.**for** ii=1:size(PCA_Ele,2) **do**    FuseInput(:,(ii−1)×2+1)=PCA_Ele(:,ii);    FuseInput(:,2×ii)=PCA_Azi(:,ii);    FuseInput(:,2×size(PCA_Ele,2)+1)=Label;**end for**RangeFusion=CNN-BiLSTM(FuseInput,KFoldNum);

**Algorithm 4** TFFusion()
**Input:** Elevation, Azimuth and Label of Sample Data, KFoldNum, Frebin, TFWin
**Output:** Fusion Feature of TF image of Samples
$Ele=\mathrm{tfrmhs}(Elevation^{\prime },1:\mathrm{length}\left(Elevation\right),Frebin,TFWin)$;
$Azi=\mathrm{tfrmhs}(Azimuth^{\prime },1:\mathrm{length}\left(Azimuth\right),Frebin,TFWin)$;
Calculate PCANet of Range Profiles of Samples via SVD.
**for** ii=1:size(PCA_Ele,2) **do**
    FuseInput(:,(ii−1)×2+1)=PCA_Ele(:,ii);
    FuseInput(:,2×ii)=PCA_Azi(:,ii);
    FuseInput(:,2×size(PCA_Ele,2)+1)=Label;

**end for**

TFFusion=CNN-BiLSTM(FuseInput,KFoldNum);


The third one of the key aspects of our method is fusing the PCANet of 3D range–azimuth–time. Because the actions are temporal processes, fusing 3D range–azimuth–time temporally first can achieve higher quality features. The 3D range-azimuth-time fusion procedure was carried out on the CNN-BiLSTM structure with temporal fusion first, and PCANet fusion followed to ensure the best fusion, as shown in Algorithm 5. In this part, we calculated the range–azimuth–time using elevation and azimuth data instead of the angle relation fused I/Q data.
**Algorithm 5** RanAziTimeFusion()**Input:** Elevation, Azimuth and Label of Sample Data, KFoldNum**Output:** Fusion Feature of Range–Azimuth–TimeCalculate Range–Azimuth–Time of Elevation and Azimuth, respectively.Calculate the PCANet of Range–Azimuth–Time.%PCAofRAT(SampleNum,FrameNum,PCANum)**for** ii=1 to FrameNum **do**    Fusion(:,(ii−1)∗2+1,:)=PCAofRAT_Ele(:,ii,:)    Fusion(:,2∗ii,:)=PCAofRAT_Azi(:,ii,:)**end for****for** ii=1 to PCANum **do**    Input(:,:)=Fusion(:,:,ii)    Temp=[Input,Label]    FuseTime=CNN-BiLSTM(Temp,KFoldNum)**end for**FuseInput=[FuseTime,Label]RATFusion=CNN-BiLSTM(FuseInput,KFoldNum)

## 4. Experimental Results and Analysis

This section will display the experiment setup and data collection in the first part and the implementation details and performance analysis in the following parts.

### 4.1. Experiment Setup and Data Collection

Classifying actions by individual subjects is straightforward, but identifying combinations of actions presents challenges. The experiments in this paper aimed to classify these action combinations. In this article, two evaluation modules named AWR2243 and DCA1000 EVM were applied for the experiment. The AWR2243 is the ideal mmWave radar sensor because of its ease of use, low cost, low power consumption, high-resolution sensing, precision, and compact size, whose parameters are listed in Table 3.

For the experimental setting, an activity room (6 m × 8 m) within the Doctoral Building of the University of Glasgow was chosen to support ample ground for our experiments. The distances between the testee (1.5 m), glass wall, and AWR2243 are shown in Figure 8. The radar resolution of the azimuth and elevation were $15^{\circ }$ and $30^{\circ }$, respectively. In the experiment, the azimuth angle was mainly concerned with the moving testee. Therefore, the angular separation should range from $20^{\circ }$ to $40^{\circ }$, which can resolve the two subjects spatially. The horizontal and vertical radar fields of view (FoVs) are both $60^{\circ }$, giving the radar FoV a concical shape. In our experiments, all testing subjects were within the radar’s FoV. However, if additional targets are present outside the FoV, they may appear as ‘ghost’ targets in the radar spectrogram. The further these targets are from the radar’s FoV, the lower their signal-to-noise ratio (SNR) will be.

Nine males and nine females participated in the experiment, with participants’ identities made confidential for privacy. Detailed participant information is available in Table 4.

The experimental data were collected from fourteen categories of combinations of actions such as (I) bend and bend, (II) squat and bend, (III) stand and bend, (IV) walk and bend, (V) fall and bend, (VI) squat and squat, (VII) stand and squat, (VIII) fall and squat, (IX) walk and squat, (X) stand and stand, (XI) walk and stand, (XII) fall and stand, (XIII) walk and walk, and (XIV) fall and walk. There were 1500 samples in every category, with 21,000 samples in total. The time length of every sample was 3 s, with 800 Hz sampling after desampling processing. Hence, 21,000 was the total number of 3 s samples, while 2400 was the total number of frames of every sample, with the scatter plot shown in Figure 9. Moreover, the coherent processing interval (CPI) was 80 ms.

### 4.2. Implement Details

In this section, we applied five statistics (sensitivity, precision, F1, specificity, and accuracy) to measure the performance. These measures can be expressed as
(9)$$Sensitivity=\frac{TP}{TP+FN},$$
(10)$$Precision=\frac{TP}{TP+FP},$$
(11)$$F1=\frac{2TP}{2TP+FP+FN},$$
(12)$$Specificity=\frac{TN}{TN+FP},$$
(13)$$Accuracy=\frac{TP+TN}{TP+FP+TN+FN},$$
where TP and FP are the number of true and false positives, while TN and FN are the number of true and false negatives. A true positive means a combination of actions is labeled correctly, a false positive means another combination of actions is labeled as the combination of actions, a true negative is a correct rejection, and a false negative is a missed detection.

Moreover, besides the STFT, three additional time–frequency distributions, named Margenau–Hill Spectrogram, Choi–Williams, and smoothed pseudo Wigner–Ville, contributed in the final recognition and will be compared in the following part. The four expressions of time–frequency can be written as (14)$$\begin{array}{cc} {\mathrm{STFT}}_{x}\left(t,f;h\right) & ={\;\int \int \;}_{-\infty }^{+\infty }x(u)H^{*}(u-t)e^{\left(-j2\pi fu\right)}du \end{array},$$
(15)$$\begin{array}{cc} {\mathrm{MHS}}_{x}\left(t,f\right) & =R\{K_{\mathrm{gh}}^{-1}{\mathrm{STFT}}_{x}\left(t,f;g\right){\mathrm{STFT}}_{x}^{*}\left(t,f;h\right)\}\;\\ K_{\mathrm{gh}} & =\int h\left(u\right)g^{*}\left(u\right)du \end{array},$$
(16)$$\begin{array}{cc} {\mathrm{SPWV}}_{x}\left(t,\tau \right)= & {\;\int \int \;}_{-\infty }^{+\infty }h\left(\tau \right){\;\int \int \;}_{-\infty }^{+\infty }g\left(s-t\right)\\ \; & x\left(s+\frac{\tau }{2}\right)x^{*}\left(s-\frac{\tau }{2}\right)dse^{-j2\pi v\tau }d\tau \end{array},$$
(17)$$\begin{array}{cc} {\mathrm{CW}}_{x}\left(t,f\right)= & 2{\;\int \int \;}_{-\infty }^{+\infty }\frac{\sqrt{\sigma }}{4\sqrt{\pi }\left|\tau \right|}e^{-\frac{f^{2}\sigma }{16\tau^{2}}}\\ \; & x\left(t+f+\frac{\tau }{2}\right)x^{*}\left(t+f-\frac{\tau }{2}\right)e^{-i2\pi \tau }dfdt \end{array},$$
where x is the input signal, t represents the vectors of time instants, and f represents the normalized frequencies. σ denotes the square root of the variance. ${\mathrm{STFT}}_{x}\left(\cdot \right)$ is the short-time Fourier transform of x. h is a smoothing window, while g is the analysis window.

In the feature extraction step, 5-fold cross-validation was employed to split the dataset into training and testing sets. Following the computation of five sets of fusion features, the results from these five groups were reconstructed into new datasets. Ultimately, the final classification outcomes for these new datasets were determined based on 10-fold cross-validation.

### 4.3. Performance Analysis

Figure 10 draws the classification sensitivity of every combination of actions of competitive temporal neural networks, such as the gated recurrent unit (GRU), LSTM, residual and LSTM recurrent networks (MSRLSTM), BiLSTM, attention-based BiLSTM (ABLSTM), temporal pyramid recurrent neural network (TPN), temporal CNN (TCN), and CNN-BiLSTM. The option parameters of these temporal neural network algorithms were all six hidden units, with Adam as the optimizer during training, a 0.0001 ℓ2 regularization, a 0.001 initial learn rate, a 0.5 drop factor learning rate, and 100 epochs. The time length of every sample was 3 s with 0.00125 s as the time interval. All the results are based on the average of 10-fold cross-validation. The average classification sensitivity for these action combinations is as follows: 34.56% for GRU, 35.57% for LSTM, 13.05% for MSRLSTM, 34.11% for BiLSTM, 34.95% for ABLSTM, 7.18% for TPN, 26.91% for TCN, and 33.69% for CNN-BiLSTM. All the algorithms in this figure serve as comparison methods for our proposed approach. Moveover, although CNN-BiLSTM did not have the highest average performance in temporal signal classification, we selected CNN-BiLSTM as the fusion method due to its data compatibility with the 3D fusion requirement in our algorithm.

As shown in Figure 7, the extracted features in our method came from the statistical offset parameters, range profiles, time–frequency, and azimuth–range–time. The offset parameters were based on the statistical calculation of the signal fused according to its elevation and azimuth information. The feature of range profiles was the output of the fusion of the PCANet of range profiles. The time–frequency feature was the output of the fusion of the PCANet of time–frequency images. The TF toolbox [49] analyzed the time–frequency, including the STFT, Margenau–Hill Spectrogram, Choi–Williams, and smoothed pseudo Wigner–Ville, i.e., (tfrstft.m, tfrmhs.m, tfrcw.m, and tfrspwv.m). The Frequency bin was 128, with a 127 hamming window. The feature of azimuth–range–time is the fusion output of both the temporal and PCANet of azimuth–range–time. Besides the offset parameters calculation, all the other feature fusion methods are based on CNN-BiLSTM. Moreover, all output results from CNN-BiLSTM in this paper are based on the 5-fold cross-validation to obtain the fusion output. Then, nine fused feature vectors were obtained as the classifier’s inputs.

To find out the most suitable classifier for our approach, we utilized eight classifiers for the final classification, such as naïve Bayes (NB), pseudo quadratic discriminant analysis (PQDA), diagonal quadratic discriminant analysis (DQDA), K-nearest neighbors (K = 3), boosting, bagging, random forest (RF), and support vector machine (SVM). The SVM had the kernel of a two-order polynomial function with the auto-kernel scale, and the constraint was set to one with true standardization. The time length of every sample was 3 s, with 800 Hz sampling after desampling processing. Every group of 10-fold cross-validation was used by selecting 90% (18,900 samples) for learning features and the remaining 10% (2100 samples) for testing. All the results in this article are based on the average of all these 10 folds.

Figure 11 shows our average sensitivity of fourteen categories via various classifiers under the time–frequency condition of the STFT, Margenau–Hill Spectrogram, Choi–Williams, smoothed pseudo Wigner–Ville, and the joint of the above four methods, whose details are shown in Table 5. In CNN-BiLSTM processing, the epoch number was 100, and the fusion output was based on the average of five groups of 5-fold cross-validation. The final classification performance of all trials from the 10-fold partitions was the average of all ten groups. In this figure, the sensitivities from the bagging, random forest, and SVM surpassed 94%, while random forest, as the classifier, played better than the others. Thereby, random forest was subsequently considered as the unique classifier in the following research for convenience.

To evaluate the effect of the number of epochs of CNN-BiLSTM structure on the different time–frequency features for final classification performance, we tested the feature performance of our method under different epoch numbers with random forest as the classifier. The accuracies of these tests are displayed in Figure 12. In this figure, the joint of the four methods performed the best, followed by MHS. The vector number of the time–frequency feature was four, while that of the others was only one. The performance differences between the joint of four methods and MHS with 100 epochs came out to 0.15% sensitivity, 0.016% F1, 0.016% precision, 0.01% specificity, and 0.02% accuracy with 100 epochs. Hence, the time–frequency analysis method recommended in this article was MHS for fairness, which will be displayed in the following research of this paper for convenience.

Figure 13 displays the precision for classification using random forest as the classifier and MHS as the time–frequency analysis method. Details of this setup are listed in Table 6. The features in Test 1 and Test 2 include six offset parameters, 1 or 10 PCA values per range profile, and MHS images. In Test 3, the features consist of six offset parameters, the most significant PCA value of the range profile image, and the PCANet fusion of the MHS image. The key difference between Test 1 and Test 3 is whether the most significant PCA value or the fused PCANet of the MHS image was used. In Test 4, the features include six offset parameters and PCANet fusions of both the range profile and MHS images. The difference between Test 3 and Test 4 is whether the feature set used the most significant PCA value or the PCANet fusion of the range profile images. Test 5 features include PCANet fusion features derived from range profiles, time–frequency analyses, and azimuth–range–time imagery. The difference between Test 4 and Test 6 (our method) is whether the feature vector came from the range–azimuth–time data. The difference between Test 5 and Test 6 is whether the feature vectors were derived from the offsets.

In Table 6, it is shown that while the feature quantity could improve final classification performance, the feature quality had a greater impact on the boosting classification performance compared to the feature quantity. Table 7 presents the classification performance of random forest for each individual feature type from Table 6. Comparing the performance of Test 4 and Test 1, the features based on image PCANet fusion improved the final results by 10.92% in sensitivity, 11.01% in precision, 11.01% in F1, 0.84% in specificity, and 1.56% in accuracy. In Test 6, the offset feature vectors improved sensitivity, precision, F1, specificity, and accuracy by 1.34%, 1.34%, 1.34%, 0.10%, and 0.19%, respectively, compared to Test 5. Additionally, features derived from range–azimuth–time via temporal and PCANet fusion boosted sensitivity, precision, F1, specificity, and accuracy by 2.56%, 2.53%, 2.55%, 0.20%, and 0.37%, respectively, compared to Test 4. Table 8 and Table 9 provide the confusion matrix and performance metrics for every action combination in Test 6.

## 5. Conclusions and Future Works

This paper introduces a pioneering method that utilizes statistical offset measures, range profiles, time–frequency analyses, and azimuth–range–time evaluations to effectively categorize various human daily activities. Our approach employs nine feature vectors, encompassing six statistical offset measures (mean, standard deviation, variance, skewness, kurtosis, and second-order central moments) and three principal component analysis network (PCANet) fusion attributes. These statistical measures were derived from combined elevation and azimuth data, considering their spatial angle connections. Fusion processes for range profiles, time–frequency analyses, and 3D range–azimuth–time evaluations were executed through concurrent 1D CNN-BiLSTM networks, with a focus on temporal integration followed by PCANet application.

The effectiveness of this approach was demonstrated through rigorous testing, showcasing enhanced robustness particularly when employing the Margenau–Hill Spectrogram (MHS) for time–frequency analysis across fourteen distinct categories of human actions, including various combinations of bending, squatting, standing, walking, and falling. When tested with the random forest classifier, our method outperformed other classifiers in terms of overall efficacy, achieving impressive results: an average sensitivity, precision, F1, specificity, and accuracy of 98.25%, 98.25%, 98.25%, 99.87%, and 99.75%, respectively.

Research in radar-based human activity recognition has made strides, yet several promising areas remain in their infancy. Future directions include the following:

Radio-frequency-based activity recognition is known for minimizing privacy intrusions compared to traditional methods. Unlike camera-based systems that produce clear images through the combination of phase and amplitude in radio frequency signals, radio-frequency-based methods still raise privacy concerns, particularly when features could be linked to personal habits, or the handling of personal data requires careful consideration. An emerging field of study focuses on safeguarding user privacy while accurately detecting their activities.

For practical application, it is crucial to deploy activity recognition models in real time. This involves segmenting received signals into smaller sections for analysis. Addressing variables like window size and overlap during segmentation is vital. Moreover, training classification models presents unique challenges, especially in recognizing transitions between activities. Labeling these transitions and employing AI algorithms to fine-tune segmentation parameters represents a significant area for development.

Data augmentation has proven useful in image classification via generative adversarial networks (GANs). Applying GANs for augmenting time series data involves converting these data into images for augmentation and then back into time series format, addressing the issue of insufficient training data for classifiers—especially deep neural networks. Moreover, accurately labeling collected data without compromising privacy is challenging. While cameras are commonly used to verify data labels, they pose privacy risks, making unsupervised learning approaches that can cluster and label data without supervision increasingly relevant.

Moreover, the validation of our proposed method on other public radar datasets will be part of future work.

## Figures and Tables

**Figure 1 sensors-24-05450-f001:**
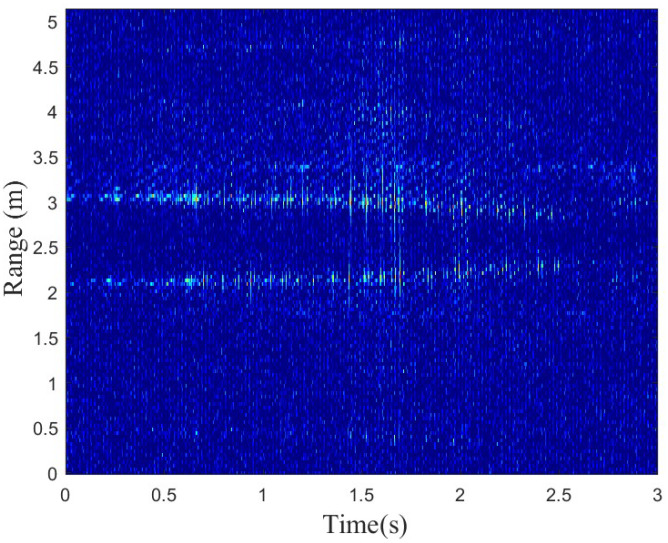
This figure shows the range profiles of measured data with two subjects doing standing (lower) and falling (upper).

**Figure 2 sensors-24-05450-f002:**
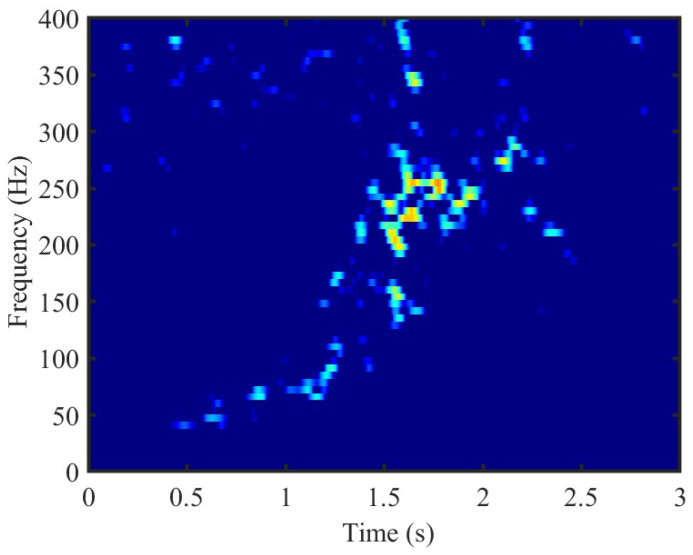
This figure displays the STFT image of measured data in Figure 1 with 800 Hz sampling frequency.

**Figure 3 sensors-24-05450-f003:**
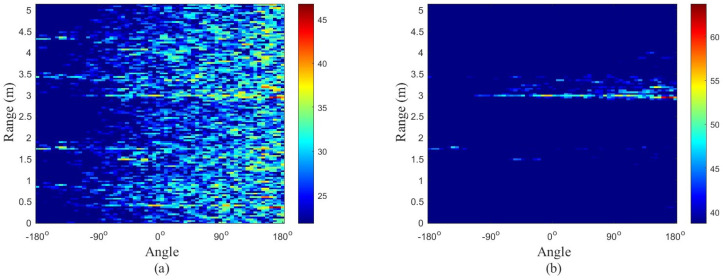
These images display the relationships between (**a**) horizontal and (**b**) vertical range and angles using elevation and azimuth information of measured data in Figure 1.

**Figure 4 sensors-24-05450-f004:**
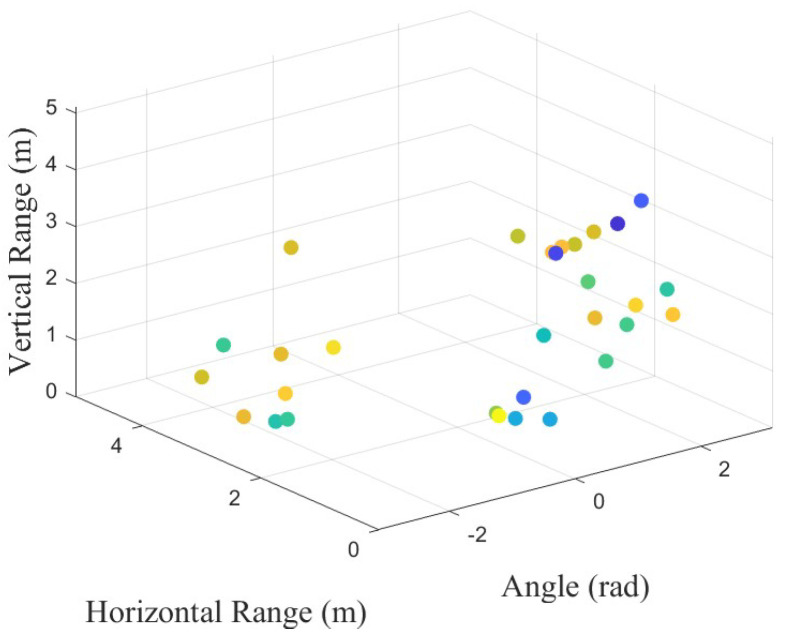
This figure shows the instantaneous 3D point cloud of measured data in Figure 3.

**Figure 5 sensors-24-05450-f005:**
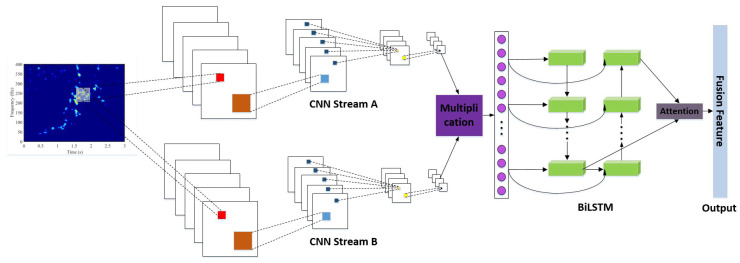
This figure shows the CNN-BiLSTM structure.

**Figure 6 sensors-24-05450-f006:**
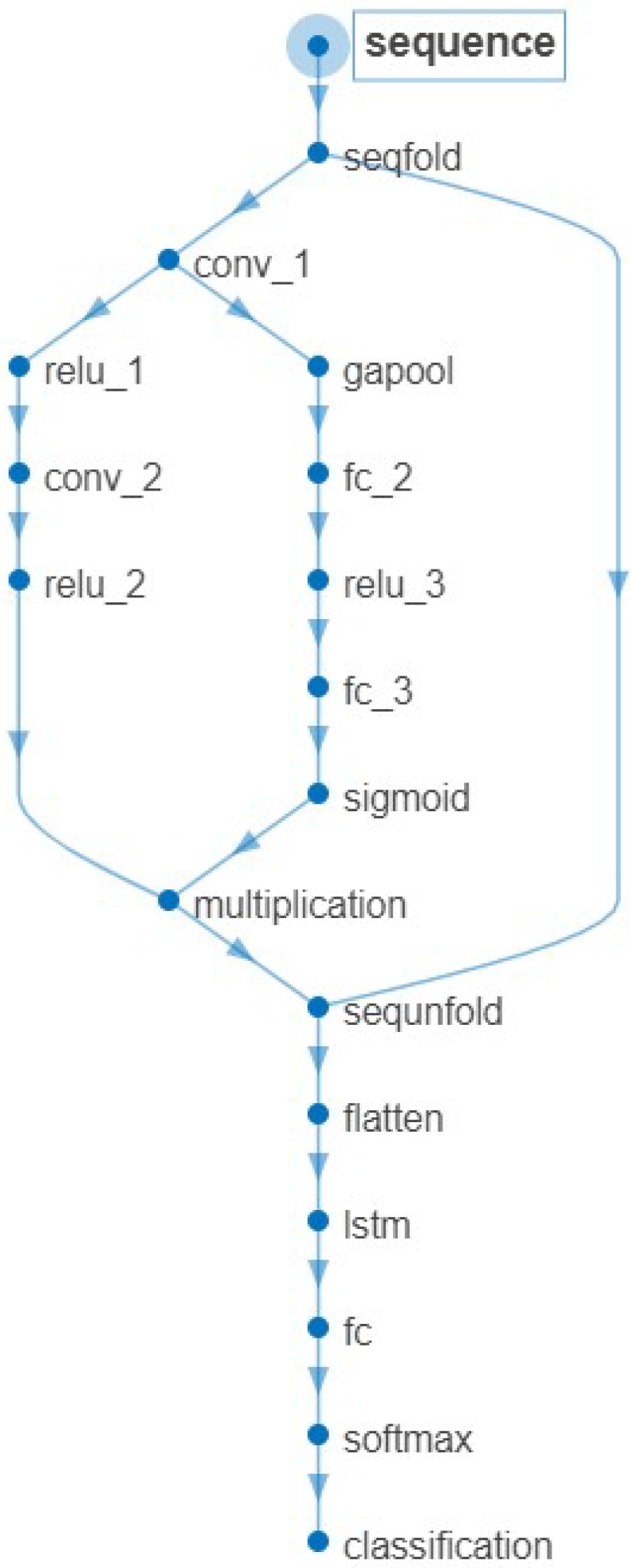
This figure displays the trainNetwork.

**Figure 7 sensors-24-05450-f007:**
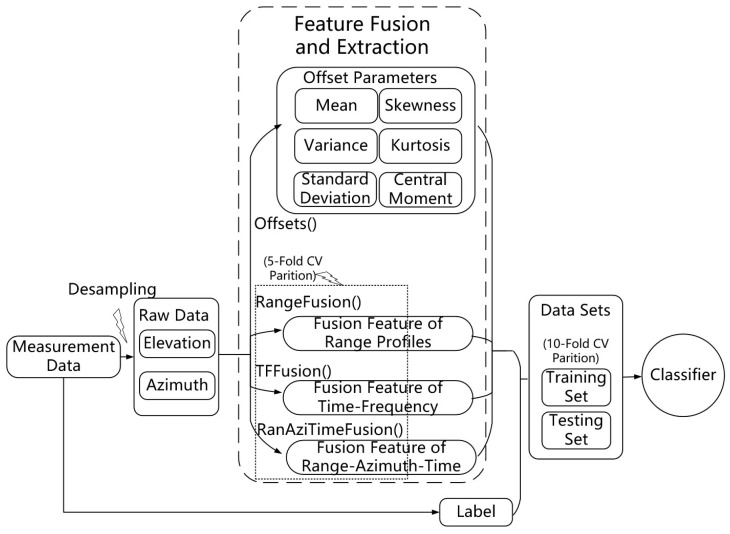
This figure shows the framework of our method.

**Figure 8 sensors-24-05450-f008:**
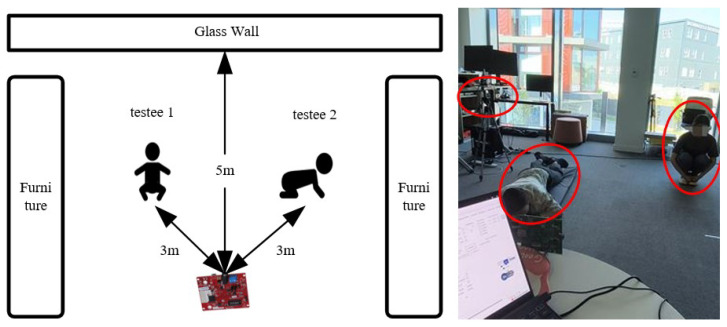
This figure displays the experimental scene diagram and photo.

**Figure 9 sensors-24-05450-f009:**
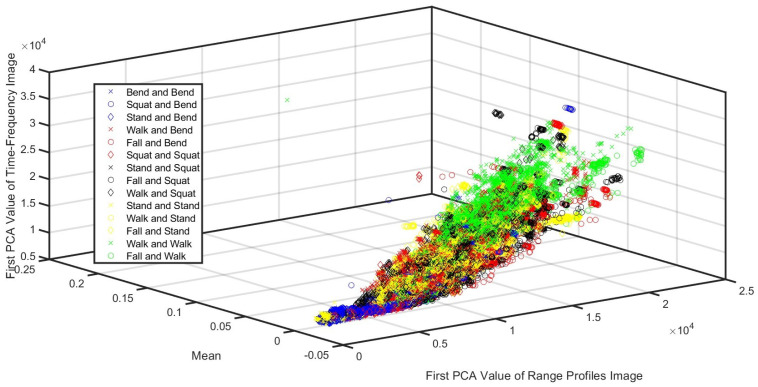
This figure shows the scatter plot of the fourteen categories of samples.

**Figure 10 sensors-24-05450-f010:**
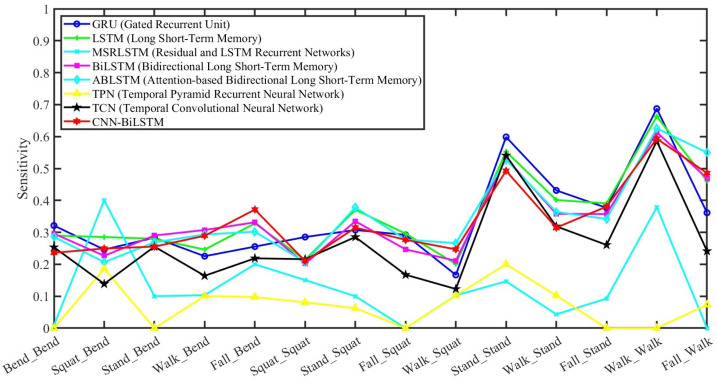
This figure shows the classification sensitivity of these combinations of actions using competitive temporal neural networks.

**Figure 11 sensors-24-05450-f011:**
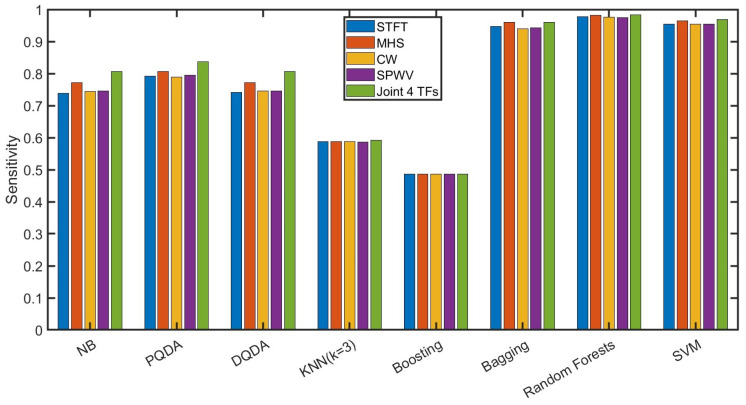
This figure shows our average sensitivity of fourteen categories via various classifiers under the time–frequency condition of STFT, Margenau–Hill Spectrogram, Choi–Williams, smoothed pseudo Wigner–Ville, and the joint of the above four methods.

**Figure 12 sensors-24-05450-f012:**
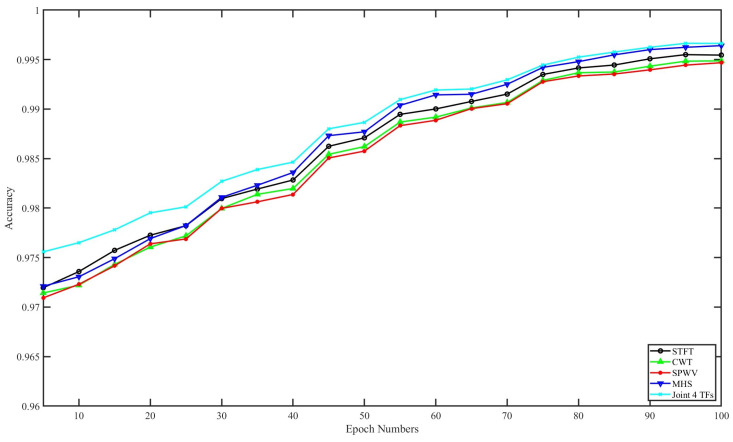
This figure shows the accuracy of our method under different epoch numbers.

**Figure 13 sensors-24-05450-f013:**
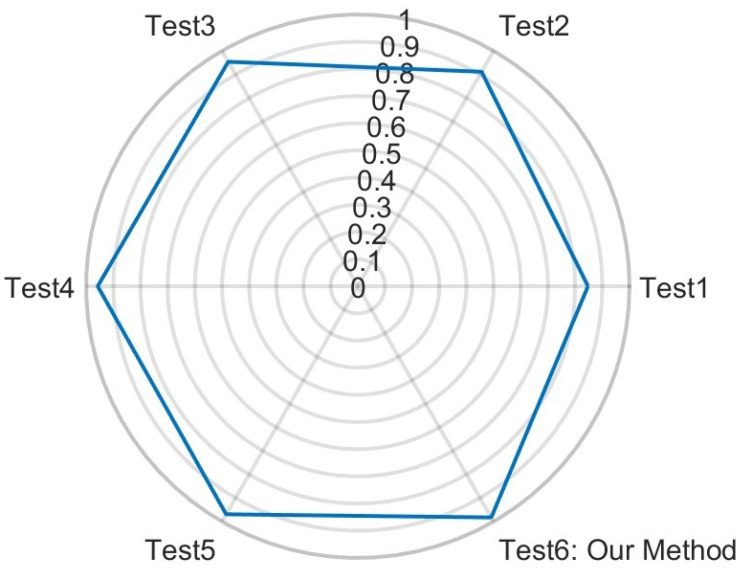
This figure shows the precision of our method (Test 6) and alternative method (Test 1–5).

**Table 1 sensors-24-05450-t001:** Option parameters of CNN-BiLSTM.

Optimizer	Parameters
Optimization	adam
Initial Learn Rate	0.001
$\ell_{2}$ regularization	1 × 10${\;\;}^{-4}$
Learn Rate Schedule	piecewise
Learn Rate Drop Factor	0.5
Learn Rate Drop Period	400
Shuffle	every epoch

**Table 2 sensors-24-05450-t002:** Details of trainNetwork Usage in Figure 6.

Name	Type	Activations	Learnable Properties
**sequence**	Sequence Input	256(S) × 1(S) × 1(C) × 1(B) × 1(T)	-
**seqfold**	Sequence Folding	out 256(S) × 1(S) × 1(C) × 1(B)	-
		miniBatchSize 1(C) × 1(U)	
**conv_1**	2D Convolution	254(S) × 1(S) × 32(C) × 1(B)	Weights 3 × 1 × 1 × 32
			Bias 1 × 1 × 32
**relu_1**	ReLU	254(S) × 1(S) × 32(C) × 1(B)	-
**conv_2**	2D Convolution	252(S) × 1(S) × 64(C) × 1(B)	Weights 3 × 1 × 32 × 64
			Bias 1 × 1 × 64
**relu_2**	ReLU	252(S) × 1(S) × 64(C) × 1(B)	-
**gapool**	2D Global	1(S) × 1(S) × 32(C) × 1(B)	-
	Average Pooling		
**fc_2**	Fully Connected	1(S) × 1(S) × 16(C) × 1(B)	Weights 16 × 32
			Bias 16 × 1
**relu_3**	ReLU	1(S) × 1(S) × 16(C) × 1(B)	-
**fc_3**	Fully Connected	1(S) × 1(S) × 64(C) × 1(B)	Weights 64 × 16
			Bias 64 × 1
**sigmoid**	Sigmoid	1(S) × 1(S) × 64(C) × 1(B)	-
**multiplication**	Elementwise	252(S) × 1(S) × 64 (C) × 1(B)	-
	Multiplication		
**sequnfold**	Sequence	252(S) × 1(S) × 64(C) × 1(B) × 1(T)	-
	Unfolding		
**flatten**	Flatten	16,128(C) × 1(B) × 1(T)	-
**lstm**	BiLSTM	12(C) × 1(B)	InputWeights 48 × 16,128
			RecurrentWeights 48 × 6
			Bias 48 × 1
**fc**	Fully Connected	14(C) × 1(B)	Weights 14 × 12
			Bias 14 × 1
**softmax**	Softmax	14(C) × 1(B)	-
**classification**	Classification	14(C) × 1(B)	-
	Output		

**Table 3 sensors-24-05450-t003:** Parameters of experimental device.

Parameters	Value
Radar Name	AWR2243
Frequency Range	76–81 GHz
Carrier Frequency	77 GHz
Number of Receivers	4
Number of Transmitters	3
Number of Samples per Chirp	256
Number of Chirps per Frame	128
Bandwidth	4 GHz
Range Resolution	4 cm
ADC Sampling Rate (max)	45 Msps
MIMO Modulation Scheme	TDM
Interface Type	MIPI-CSI2, SPI
Rating	Automotive
Operating Temperature Range	−40 to 140 ${\;\;}^{\circ }$C
TI Functional Safety Category	Functional Safety Compliant
Power Supply Solution	LP87745-Q1
Evaluation Module	DCA1000

**Table 4 sensors-24-05450-t004:** Information of the participants.

Participant ID	Gender	Height (cm)	Age
Participant 1	Male	172	25
Participant 2	Male	167	23
Participant 3	Male	165	24
Participant 4	Male	177	22
Participant 5	Male	174	26
Participant 6	Male	180	23
Participant 7	Male	185	25
Participant 8	Male	183	24
Participant 9	Male	188	22
Participant 10	Female	157	25
Participant 11	Female	155	23
Participant 12	Female	159	24
Participant 13	Female	167	22
Participant 14	Female	165	26
Participant 15	Female	170	23
Participant 16	Female	175	25
Participant 17	Female	173	24
Participant 18	Female	177	22

**Table 5 sensors-24-05450-t005:** Performance details of Figure 11.

Sensitivity	STFT	MHS	CW	SPWV	Joint 4TFs
**NB**	0.7380	0.7710	0.7449	0.7460	0.8066
**PQDA**	0.7926	0.8063	0.7889	0.7950	0.8376
**DQDA**	0.7414	0.7717	0.7458	0.7454	0.8062
**KNN (k = 3)**	0.5877	0.5875	0.5880	0.5861	0.5920
**Boosting**	0.4861	0.4863	0.4858	0.4862	0.4862
**Bagging**	0.9473	0.9595	0.9401	0.9424	0.9594
**Random Forest**	0.9771	0.9825	0.9754	0.9746	0.9840
**SVM**	0.9547	0.9651	0.9545	0.9550	0.9682
**Precision**	**STFT**	**MHS**	**CW**	**SPWV**	**Joint 4TFs**
**NB**	0.7663	0.7929	0.7691	0.7692	0.8176
**PQDA**	0.8127	0.8232	0.8105	0.8163	0.8503
**DQDA**	0.7691	0.7934	0.7700	0.7699	0.8170
**KNN (k = 3)**	0.5954	0.5952	0.5958	0.5938	0.5995
**Boosting**	0.3257	0.3399	0.3647	0.3970	0.3764
**Bagging**	0.9479	0.9598	0.9409	0.9429	0.9597
**Random Forest**	0.9772	0.9825	0.9755	0.9747	0.9841
**SVM**	0.9552	0.9655	0.9549	0.9554	0.9684
**F1**	**STFT**	**MHS**	**CW**	**SPWV**	**Joint 4TFs**
**NB**	0.7058	0.7453	0.7127	0.7140	0.7884
**PQDA**	0.7701	0.7879	0.7652	0.7712	0.8235
**DQDA**	0.7101	0.7459	0.7144	0.7139	0.7873
**KNN (k = 3)**	0.5859	0.5854	0.5861	0.5843	0.5900
**Boosting**	0.3570	0.3577	0.3748	0.3765	0.3681
**Bagging**	0.9471	0.9594	0.9399	0.9422	0.9594
**Random Forest**	0.9772	0.9825	0.9754	0.9746	0.9841
**SVM**	0.9548	0.9651	0.9546	0.9551	0.9682
**Accuracy**	**STFT**	**MHS**	**CW**	**SPWV**	**Joint 4TFs**
**NB**	0.9798	0.9824	0.9804	0.9805	0.9851
**PQDA**	0.9840	0.9851	0.9838	0.9842	0.9875
**DQDA**	0.9801	0.9824	0.9804	0.9804	0.9851
**KNN (k = 3)**	0.9683	0.9683	0.9683	0.9682	0.9686
**Boosting**	0.9605	0.9605	0.9604	0.9605	0.9605
**Bagging**	0.9959	0.9969	0.9954	0.9956	0.9969
**Random Forest**	0.9982	0.9987	0.9981	0.9980	0.9988
**SVM**	0.9965	0.9973	0.9965	0.9965	0.9976
**Specificity**	**STFT**	**MHS**	**CW**	**SPWV**	**Joint 4TFs**
**NB**	0.9626	0.9673	0.9636	0.9637	0.9724
**PQDA**	0.9704	0.9723	0.9698	0.9707	0.9768
**DQDA**	0.9631	0.9674	0.9637	0.9636	0.9723
**KNN (k = 3)**	0.9411	0.9411	0.9411	0.9409	0.9417
**Boosting**	0.9266	0.9266	0.9265	0.9266	0.9266
**Bagging**	0.9925	0.9942	0.9914	0.9918	0.9942
**Random Forest**	0.9967	0.9975	0.9965	0.9964	0.9977
**SVM**	0.9935	0.9950	0.9935	0.9936	0.9955

**Table 6 sensors-24-05450-t006:** Performance fetails of Figure 13.

	Test 1	Test 2	Test 3	Test 4	Test 5	Test 6
**Statistical Offsets**	6	6	6	6	0	6
**Range Features**	1	10	1	1	1	1
**TF Features**	1	10	1	1	1	1
**Range–Azimuth–Time**	0	0	0	0	1	1
**Total Features**	8	26	8	8	3	9
**Sensitivity**	0.8477	0.9115	0.9535	0.9569	0.9691	0.9825
**Precision**	0.8472	0.9114	0.9538	0.9572	0.9692	0.9825
**F1**	0.8469	0.9112	0.9535	0.9570	0.9691	0.9825
**Specificity**	0.9883	0.9932	0.9964	0.9967	0.9976	0.9987
**Accuracy**	0.9782	0.9874	0.9934	0.9938	0.9956	0.9975

**Table 7 sensors-24-05450-t007:** Performance details of single-feature type of Table 6.

Single Feature Type	Feature Vectors	Sensitivity	Precision	F1	Specificity	Accuracy
**Mean**	1	0.0699	0.0701	0.0697	0.9285	0.8671
**Variance**	1	0.2605	0.2585	0.2588	0.9431	0.8944
**Standard Deviation**	1	0.2596	0.2576	0.2578	0.9430	0.8942
**Kurtosis**	1	0.2058	0.2055	0.2050	0.9389	0.8865
**Skewness**	1	0.1664	0.1670	0.1661	0.9359	0.8809
**Central Moment**	1	0.2485	0.2470	0.2471	0.9422	0.8926
**Offset Parameters**	6	0.2769	0.2763	0.2758	0.9444	0.8967
**PCA of Range Profiles**	1	0.2769	0.2763	0.2758	0.9444	0.8967
**PCA of Range Profiles**	10	0.7675	0.7706	0.7663	0.9821	0.9668
**PCANet Fusion of Range Profiles**	1	0.6758	0.7048	0.6707	0.9751	0.9537
**PCA of TF Image**	1	0.3555	0.3551	0.3545	0.9504	0.9079
**PCA of TF Image**	10	0.7910	0.7960	0.7907	0.9839	0.9701
**PCANet Fusion of TF Image**	1	0.8937	0.8975	0.8941	0.9918	0.9848
**Fusion of Range–Azimuth–Time**	1	0.9253	0.9286	0.9251	0.9943	0.9893

**Table 8 sensors-24-05450-t008:** Confusion matrix of our method with STFT, 100 epochs, and random forest.

Action Combinations	ID	I	II	III	IV	V	VI	VII	VIII	IX	X	XI	XII	XIII	XIV
**Bend and Bend**	**I**	**1465**	33	2	0	0	0	0	0	0	0	0	0	0	0
**Squat and Bend**	**II**	13	**1448**	18	1	1	11	4	0	0	4	0	0	0	0
**Stand and Bend**	**III**	1	21	**1468**	0	0	6	4	0	0	0	0	0	0	0
**Walk and Bend**	**IV**	0	0	0	**1475**	15	0	0	2	5	0	3	0	0	0
**Fall and Bend**	**V**	0	0	0	4	**1470**	19	0	6	0	0	1	0	0	0
**Squat and Squat**	**VI**	1	9	1	1	12	**1450**	15	7	0	4	0	0	0	0
**Stand and Squat**	**VII**	0	1	1	0	0	16	**1475**	4	0	3	0	0	0	0
**Fall and Squat**	**VIII**	0	1	0	0	6	6	17	**1455**	9	1	3	2	0	0
**Walk and Squat**	**IX**	0	0	0	4	1	0	0	20	**1469**	1	4	1	0	0
**Stand and Stand**	**X**	0	1	1	0	0	1	1	2	0	**1494**	0	0	0	0
**Walk and Stand**	**XI**	0	0	0	0	0	0	0	2	9	0	**1485**	4	0	0
**Fall and Stand**	**XII**	0	0	0	0	0	1	1	4	1	1	4	**1488**	0	0
**Walk and Walk**	**XIII**	0	0	0	0	0	0	0	0	0	0	0	3	**1497**	0
**Fall and Walk**	**XIV**	0	0	0	0	0	0	0	0	0	0	0	5	2	**1493**

**Table 9 sensors-24-05450-t009:** The performance of the confusion matrix in Table 8.

Action Combinations	Sensitivity	Precision	F1	Specificity	Accuracy
**Bend and Bend**	0.9767	0.9899	0.9832	0.9992	0.9976
**Squat and Bend**	0.9653	0.9564	0.9608	0.9966	0.9944
**Stand and Bend**	0.9787	0.9846	0.9816	0.9988	0.9974
**Walk and Bend**	0.9833	0.9933	0.9883	0.9995	0.9983
**Fall and Bend**	0.9800	0.9767	0.9784	0.9982	0.9969
**Squat and Squat**	0.9667	0.9603	0.9635	0.9969	0.9948
**Stand and Squat**	0.9833	0.9723	0.9778	0.9978	0.9968
**Fall and Squat**	0.9700	0.9687	0.9694	0.9976	0.9956
**Walk and Squat**	0.9793	0.9839	0.9816	0.9988	0.9974
**Stand and Stand**	0.9960	0.9907	0.9934	0.9993	0.9990
**Walk and Stand**	0.9900	0.9900	0.9900	0.9992	0.9986
**Fall and Stand**	0.9920	0.9900	0.9910	0.9992	0.9987
**Walk and Walk**	0.9980	0.9987	0.9983	0.9999	0.9998
**Fall and Walk**	0.9953	1.0000	0.9977	1.0000	0.9997
**Average Performance**	0.9825	0.9825	0.9825	0.9987	0.9975

## Data Availability

Data are contained within the article.

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
