# Peer review of "Human Multi-Activities Classification Using mmWave Radar: Feature Fusion in Time-Domain and PCANet"

_sensors, 2024, doi:10.3390/s24165450_

Round 1

Reviewer 1 Report

Comments and Suggestions for Authors

In this article, a system exploiting a mmWave radar is used to detect human activities. The different activities are classified by means of a feature fusion in the time domain and PCANet via CNN-BiLSTM

The introduction is well written. It clearly states the purpose of the work and the contribution to the scientific literature.

The present article exploits different radar features, from range profile to Doppler analysis. It reminds me this recent article that the authors can consider in section 2-RelatedWork.

E. Cardillo, C. Li and A. Caddemi, "Heating, Ventilation, and Air Conditioning Control by Range-Doppler and Micro-Doppler Radar Sensor," 2021 18th European Radar Conference (EuRAD), London, United Kingdom, 2022, pp. 21-24, doi: 10.23919/EuRAD50154.2022.9784461.

In the literature, other articles describing the monitoring of the sitting-time with radar are very similar and based on similar considerations.

From the sentence “The first one introduces the formula of mmWave radar [32] used in this article.”, pag. 3, line 36, it seems that the formula is particularly innovative. However, it seems quite standard, the authors should rephrase the sentence.

The content of Fig. 1, i.e., two subjects doing standing and falling, is reported only in the figure caption but should be described in the text. Moreover, it is not possible to understand which is the standing and the falling subject.

The text font of the sequence (Fig.6) should be increased.

The text font of the Fig.7 should be increased.

What is the radar field of view? How does the FoV impact the measurement if additional targets are present outside the FoV (they can appear as ghost targets)?

Reviewer 2 Report

Comments and Suggestions for Authors

Round 2

Reviewer 2 Report

Comments and Suggestions for Authors

I recommend that the manuscript be accepted for publication in its current form, with no further revisions required.